# Comparative Analysis of Photon Stereotactic Radiotherapy and Carbon-Ion Radiotherapy for Elderly Patients with Stage I Non-Small-Cell Lung Cancer: A Multicenter Retrospective Study

**DOI:** 10.3390/cancers15143633

**Published:** 2023-07-15

**Authors:** Shuri Aoki, Hiroshi Onishi, Masataka Karube, Naoyoshi Yamamoto, Hideomi Yamashita, Yoshiyuki Shioyama, Yasuo Matsumoto, Yukinori Matsuo, Akifumi Miyakawa, Haruo Matsushita, Hitoshi Ishikawa

**Affiliations:** 1QST Hospital, National Institutes for Quantum Science and Technology, Chiba 263-8555, Japan; n.yamamoto@chouseihp.jp (N.Y.); ishikawa.hitoshi@qst.go.jp (H.I.); 2Department of Radiology, University of Tokyo Hospital, Tokyo 113-8655, Japan; yamashitah-rad@h.u-tokyo.ac.jp; 3Department of Radiology, University of Yamanashi, Yamanashi 400-0016, Japan; honishi@yamanashi.ac.jp; 4Department of Radiology, Teikyo University Mizonokuchi Hospital, Kanagawa 213-8507, Japan; karube.masataka.jd@teikyo-u.ac.jp; 5Ion Beam Therapy Center, SAGA HIMAT Foundation, Saga 841-0071, Japan; shioyama-yoshiyuki@saga-himat.jp; 6Department of Radiation Oncology, Niigata Cancer Center Hospital, Niigata 951-8133, Japan; ymatsu@niigata-cc.jp; 7Department of Radiation Oncology and Image-Applied Therapy, Graduate School of Medicine, Kyoto University, Kyoto 606-8501, Japan; ymatsuo@med.kindai.ac.jp; 8Department of Radiation Oncology, Faculty of Medicine, Kindai University, Osaka 577-8502, Japan; 9Department of Radiology, Graduate School of Medical Sciences, Nagoya City University, Aichi 467-8501, Japan; miyakawa.akifumi.sj@mail.hosp.go.jp; 10Department of Radiation Oncology, Graduate School of Medicine, Tohoku University, Sendai 980-8577, Japan; haruo.matsushita.c8@tohoku.ac.jp

**Keywords:** lung neoplasms, elderly, carbon-ion radiotherapy, stereotactic body radiotherapy

## Abstract

**Simple Summary:**

Surgery is the standard treatment for stage I non-small-cell lung cancer (NSCLC), but is often difficult to perform in elderly patients due to their comorbidities and inferior physical status. Stereotactic body radiotherapy (SBRT) is expected to be an alternative treatment and is becoming widely used. Meanwhile, carbon ion radiotherapy (CIRT) is being recognized as an attractive treatment option with potentially lower toxicity for elderly patients with co-morbidities. The aim of this retrospective study was to assess the efficacy and safety of the two modalities for NSCLC in the elderly using Japanese multicenter data. The data on SBRT and single CIRT in consecutive patients ≥80 years of age suggested that the two modalities were comparable in terms of efficacy, at least in Japan during the study period; single CIRT was associated with a lower incidence of severe radiation pneumonitis, indicating a potentially safer treatment.

**Abstract:**

The emergence of an aging society and technological advances have made radiotherapy, especially stereotactic body radiotherapy (SBRT), a common alternative to surgery for elderly patients with early stage non-small-cell lung cancer (NSCLC). Carbon-ion radiotherapy (CIRT) is also an attractive treatment option with potentially lower toxicity for elderly patients with comorbidities. We compared the clinical outcomes of the two modalities using Japanese multicenter data. SBRT (n = 420) and single-fraction CIRT (n = 70) data for patients with stage I NSCLC from 20 centers were retrospectively analyzed. Contiguous patients ≥ 80 years of age were enrolled, and overall survival (OS), disease-specific survival (DSS), local control (LC), and adverse event rates were compared. The median age was 83 years in both groups and the median follow-up periods were 28.5 and 42.7 months for SBRT and CIRT, respectively. The 3-year OS, DSS, and LC rates were 76.0% vs. 72.3% (*p* = 0.21), 87.5% vs. 81.6% (*p* = 0.46), and 79.2% vs. 78.2% (*p* = 0.87), respectively, for the SBRT vs. CIRT groups. Regarding toxicity, 2.9% of the SBRT group developed grade ≥ 3 radiation pneumonitis, whereas none of the CIRT group developed grade ≥ 2 radiation pneumonitis. SBRT and CIRT in elderly patients showed similar survival and LC rates, although CIRT was associated with less severe radiation pneumonitis.

## 1. Introduction

Lung cancer is the leading cause of cancer-related incidence and mortality in the world [1,2]. Surgery is recommended as the standard treatment for stage I non-small-cell lung cancer (NSCLC), even in the elderly population [3,4]. Age-related mortality associated with pulmonary surgery increases with age [5,6].

Elderly people tend to refuse or are not offered surgery because of their comorbidities and inferior physical status compared to those of younger patients. Japan is one of the most prevalently aging countries in the world, and it is time to seriously consider treating elderly people. In addition, the growing need for minimally invasive treatments and the development of radiotherapy technology have resulted in stereotactic body radiation therapy (SBRT), with photon beams being recognized as a promising alternative treatment for inoperable stage I NSCLC. The efficacy and safety of SBRT have been reported in several clinical trials conducted globally [7,8,9], and SBRT is widely practiced in Japan. On the other hand, carbon-ion beam radiation therapy (CIRT) is a radiation therapy (RT) modality with features of superior dose concentration and higher biological efficacy, and has been shown to be effective for stage I NSCLC [10,11,12,13].

Here, we collected data from two groups of elderly patients who underwent two forms of definitive radiotherapy for stage I NSCLC during the same period: SBRT or CIRT. This study assessed the efficacy and safety of these two modalities and explored the differences between them.

## 2. Materials and Methods

### 2.1. Data Sources

The multi-institutional data on patients aged ≥80 years treated for stage I NSCLC between April 2003 and March 2012 were retrospectively reviewed; 19 institutes provided data on photon SBRT, while data on CIRT were collected at a single institute (QST Hospital). All institutions were required to obtain approval from their institutional review boards. The study was conducted in accordance with the guidelines of the Declaration of Helsinki and approved by the hospital’s Ethics Committee (protocol code: 16–021; date of approval: 16 August 2016).

### 2.2. Eligibility Criteria

We enrolled patients with histologically or cytologically confirmed stage I NSCLC (UICC, 6th edition). All patients met the following criteria: (1) age of 80 years or older on the first day of irradiation, (2) World Health Organization performance status of 0 to 2, (3) refusal of surgery or inoperable tumors, (4) no history of radiation therapy in the lung, (5) no active infection in the irradiation field, (6) no obvious interstitial pneumonia on chest radiography, and (7) no other active cancers.

### 2.3. Radiation Therapy

Photon SBRT techniques and dose calculation methods differed between institutions. However, they had at least the following five items in common, fulfilling the principle of the “stereotactic radiotherapy technique”: (1) the image-guided technique, which was used every fraction to reproduce the position accurately; (2) a respiratory motion suppresser, which was used to ensure respiratory motion to within 5 mm; (3) computed tomography (CT) slices, which were thicker than 3 mm for three-dimensional (3D) treatment planning; (4) non-coplanar multiple beams or arcs; (5) a prescribed dose of over 5 Gy per fraction; and (6) the biological effective dose (BED) using linear-quadratic model, which was greater than 100 Gy with alpha/beta = 10 Gy.

Regarding CIRT, a dose of 28–50 Gy (relative biological effectiveness (RBE)) was delivered in a single fraction using 3D treatment planning with respiratory-gated CT images, which would be equivalent to 106.4–300 Gy (RBE) if converted into BED10 in conventional radiotherapy. The key technologies to minimize the OAR dose while covering tumor thickness were the spread Bragg peak (SOBP) of the carbon ion beam and the scanning delivery system [12,14]. Treatment planning involved the use of CT with 1–2 mm slices. Beams using 2–4 fixed ports were delivered on the same day, and the time required to irradiate 50 Gy (RBE), the dose currently used, was about 30 min if there were no major positioning problems. The dose restrictions for each OAR were as follows: V20 of <15% (less than 15% of the volume receiving 20 Gy) for the lungs’ GTV, D0.2cc (maximum dose per 0.2cc of an organ) of <10 Gy (RBE) for the trachea and esophagus, D0.2cc of <30 Gy (RBE) for the main/ lobar bronchus, and Dmax (maximum point dose for an organ) of <10 Gy (RBE). Details of the treatment have been provided in a previous report [10,11,12,13].

For target delineation in both treatments, the gross tumor volume (GTV) was contoured on the CT images showing the lung window level. The clinical target volume (CTV) was enlarged by 0–5 mm from the GTV, and an internal margin (IM) was added around the CTV at the discretion of the radiation oncologist, based on respiratory motion measurements at each institution. Gating, tracking, breath-holding, and abdominal compression techniques were used to reduce IM caused by respiratory motion. Details of the method for creating the planning target volume (PTV) or equivalent treatment target, dose prescription, and fractionation were determined at each center based on clinical considerations.

### 2.4. Endpoints and Statistics

The overall survival (OS), disease-specific survival (DSS), and local control (LC) rates were calculated from the date of initial irradiation to the date of the event or last follow-up. Each event was defined as death due to any cause, death from lung cancer or related symptoms, or recurrence within the irradiated area. All data were collected from one of the institutes (University of Yamanashi) for statistical analysis. Adverse events (AEs) were evaluated using Common Terminology Criteria for Adverse Events version 3.0. The Kaplan–Meier method was used to calculate survival rates, and *p* values were estimated using the log-rank test. The relationships between clinical factors and dosimetric parameters were evaluated using a chi-square test. Statistical analyses were performed using R software, R version 4.3.1 (https://www.r-project.org/, accessed on 30 May 2023), and the significance of the univariate analyses was set at *p* < 0.05.

## 3. Results

### 3.1. Patient and Treatment Characteristics

Comparing SBRT and CIRT, there were 420 (male:female = 282:138) and 70 (male:female = 48:22) patients treated between December 2001 and February 2016, respectively. Patient and treatment characteristics are summarized in Table 1. The median age of both groups was 83 years and there was no distribution bias. Regarding tumor size, the percentage of T2 was 33.6% for SBRT and 44.3% for CIRT, and it tended to be higher in the CIRT group, although the difference was not significant (*p* = 0.07). The number of patients with adenocarcinoma in both groups was approximately twice that of patients with squamous cell carcinoma.

The median prescribed irradiation dose was 48 Gy (range: 40–70 Gy/4–10 fractions, BED10 75–150 Gy) in the SBRT group, with a BED10 of ≥100 Gy in 91% of patients. The prescribed dose for the CIRT group was 40 Gy (RBE) (range: 28–50 Gy (RBE)).

### 3.2. Disease Control and Survival

The median follow-up was 28.5 and 42.7 months in the SBRT and CIRT groups, respectively. In the SBRT group, the 2- and 3-year OS rates were 84.0% (95% CI: 79.7–87.4%) and 76.0% (95% CI: 70.7–80.4%), respectively. In contrast, in the CIRT group, the 2- and 3-year OS rates were 84.3% (95% CI: 73.4–91.0%) and 72.3% (95% CI: 60.1–81.4%), respectively, with no significant difference between the two groups (*p* = 0.21).

The 2- and 3- year DSS rates in the SBRT group were 93.1% (95% CI: 89.8–95.4) and 87.5% (95% CI: 82.9–90.9), respectively. In the CIRT group, the 2- and 3-year DSS rates were 89.8% (95% CI: 79.7–95.0) and 81.6% (95% CI: 69.8–89.2), respectively. In addition, the corresponding 2- and 3-year rates for LC were 84.9% (95% CI: 80.5–88.4) and 79.2% (95% CI: 73.8–83.6) in the SBRT group, versus 79.9% (95% CI: 68.4–87.6) and 78.2% (95% CI: 66.4–86.2) in the CIRT group. The differences were not significant for DSS (*p* = 0.46) or LC (*p* = 0.87) (Figure 1).

### 3.3. Toxicity

Regarding radiation pneumonitis, 12 (2.4%) patients in the SBRT arm experienced grade 3 or higher pneumonitis, of which 2 (0.5%) had grade 5 pneumonitis. No other grade 3 AEs were reported.

In contrast, there were no cases of grade 2 or higher pneumonitis in the CIRT group. No other severe AEs were reported; however, of all the AEs investigated, five were evaluated as grade 2 due to musculoskeletal pain and rib fractures.

## 4. Discussion

### 4.1. Definitive Radiation Therapy for Early-Stage NSCLC in Elderly Patients

To the best of our knowledge, this is the first report to directly compare clinical data between CIRT and SBRT treatment during the same period; for CIRT, the data were available only in Japan until the time of submission of the paper. With the expected global spread of CIRT in the future, we believe that sharing the processes and challenges experienced with this treatment will be useful for its future development and appropriate use. Moreover, the appropriate management of early NSCLC in elderly patients remains controversial. Japan has one of the oldest and most rapidly aging populations worldwide, and the demand for both SBRT and CIRT among the elderly is expected to increase in the future. It is necessary to examine the safety and efficacy of SBRT and CIRT using data focusing on elderly patients, and this report is useful in this context.

SBRT has already been established as a standard treatment for stage I NSCLC, and several large multicenter prospective trials have assessed its efficacy and safety.

In a phase II clinical trial (JCOG0403) published in 2015, 165 Japanese patients with stage IA disease were treated at a dose of 48 Gy in four fractions, showing that the 3-year OS and local progression-free survival (LPFS) in the operable group were 76.5% and 68.6%, respectively [8]. Furthermore, the median age of in the study was 78 years, and the proportion of patients aged 81 years or older was 49/169 (29%), suggesting a high demand in the elderly. There have also been several previous reports on the toxicity and efficacy of SBRT in the elderly [15,16,17,18], which provide the basis for recommending SBRT to appropriately selected elderly patients. In our multicenter data, the 3-year OS and LC rates in the SBRT group were 75.8% and 78.9%, respectively. Although direct comparisons are difficult and there are variations in dose prescriptions, our results were not inferior to those of JCOG0403, a representative clinical trial conducted in Japan during the same period. The steeper gradient in survival compared to that in LC may be due to the high mortality rate in the elderly.

Toxicity was similar to or lower than that in previous SBRT reports, although those were not limited to the elderly with pneumonia above grade 3 [8,19]. However, there were two grade 5 cases, and a detailed study is required for cases in which serious AEs occurred.

CIRT, in contrast, has attracted the attention of many radiation oncologists for its theoretical benefits, both physical and biological [20], as a local treatment, although the evidence is not as well-established as that for SBRT in stage I NSCLC. Steep dose distribution is one of the features of CIRT that allows the administration of lower doses to organs at risk (OARs) while maintaining the administration of high doses to the tumor [21]. In the treatment of lung cancer, it has the advantage of preserving respiratory function and can be a salvaging therapy for elderly patients and those with interstitial pneumonia [22,23].

CIRT for NSCLC in Japan was initiated in 1994 at the QST Hospital, the CIRT site for this study. Safety and efficacy were confirmed via a gradual reduction in the number of irradiation fractions from 18 to 9 and 4, and finally via a single irradiation [9,11,24]. Subsequently, dose escalation using a single irradiation was initiated in 2005. As a result of a dose escalation trial carried out from 28 Gy (RBE) to 50 Gy (RBE), the optimal dose for stage I NSCLC was determined to be 50 Gy (RBE), the planned maximum dose in that trial, which led to the current treatment [12]. The safety of a single irradiation for elderly patients has also been reported in the same institution [25,26].

Although the CIRT group in the present study included patients who underwent dose escalation from 28 Gy (RBE) to 50 Gy (RBE), the results were comparable with those of the SBRT group in Japan during the same period. Furthermore, there were no cases of grade 2 or higher pneumonia in the CIRT group, suggesting that CIRT may be at least as safe as SBRT or even safer. CIRT is a highly recommended treatment option for elderly patients.

### 4.2. Significance of Single-Fraction CIRT

Another notable aspect of the CIRT group treatment in this study was single irradiation. The advantages of a single irradiation are the shorter treatment period and the lower stress induced on both the patient and the institution, which is especially beneficial for elderly patients. However, even for SBRT, the safety of a single irradiation has not been established, and no reports on single SBRT have been published in Japan. However, there have been a few reports from overseas. According to the results of the RTOG0915 trial published in 2019, there was no significant difference in 5-year survival or tumor control between 34 Gy in a single fraction and 48 Gy in four fractions [7]. Similarly, Singh et al. found that 30 Gy in a single fraction and 60 Gy in three fractions were equally effective [27]. The dose of a single irradiation in previous reports was mostly 28–34 Gy [7,27,28,29], and there was no significant difference between 30 Gy and 34 Gy according to a report by Videtic et al. [28]. Although the experience with single-dose irradiation is insufficient in CIRT, Yamamoto et al. reported significantly better results using 36 Gy (RBE) or higher in the dose escalation study described above, which were similar to the data for SBRT [12]. The CIRT arm of this study included cases before dose escalation; however, tumor control was not inferior to SBRT.

High-linear-energy-transfer (LET) radiation, such as CIRT, is believed to produce biological effects with fewer fractions [30,31], and the benefits of increasing the single-line volume would be significant. Although the optimal dose of CIRT has been a matter of exploration, there are several reports from the QST Hospital that 50 Gy single irradiation is safe [12,13].

An obvious advantage of a single irradiation is the reduction in the treatment period, which improves patient convenience and comfort. Several foreign studies on photon SBRT have reported an improvement in the quality of life using single irradiation [32,33]. It would also be cost-effective and would reduce the need for manpower and equipment, leading to indirect cost savings [34]. However, at present, CIRT is not covered by the Japanese insurance system, leaving unresolved issues of cost and access to widespread use.

Further, a case series of CIRT and single irradiation would clarify the patient population that would benefit the most from these therapies and allow for appropriate patient selection. Furthermore, although not directly comparable, they would provide useful data for the future consideration of shortening SBRT.

### 4.3. Limitations

This study had several major limitations. First, the patients’ backgrounds and dose prescriptions varied widely. Owing to the relatively old data, it included different dose prescriptions from the current clinical practice, with the CIRT data including cases undergoing dose escalation studies. Therefore, the efficacy and toxicity of CIRT might have been underestimated. In addition, the majority (n = 334, 80%) of the SBRT group were irradiated four times, as in JCOG 0403, but some cases with more fractions were included, indicating that lesions judged to be central were possibly included.

The second limitation was this study’s short observation period. The data in this study are insufficient to discuss long-term prognosis and late AEs. This may also have led to an overestimation of survival, which requires further research with adequate observation periods.

Third, this retrospective study was conducted at multiple institutions and different treatment protocols were used. Therefore, the insufficient clinical data made it impossible to investigate their associations with treatment outcomes. In particular, cases in which serious AEs occurred could not be discussed in detail, which remains an issue for future research.

Fourth, as mentioned above, the biological effects of photon SBRT and single-fraction CIRT cannot be directly compared, making it difficult to discuss the feasibility of single irradiation instead of photon SBRT. Proton therapy data were not included in this study. In the future, data from proton therapy, which has biological effects that are similar to those of photon therapy, should be referred to in order to determine the optimal treatment method.

Although this study has several limitations, we believe that it contains important findings regarding the future application of photon SBRT and single-fraction CIRT in Japan.

## 5. Conclusions

We retrospectively compared SBRT and single-fraction CIRT for stage I NSCLC patients aged ≥80 years. Although the limitations of the CIRT arm included insufficient low-dose treatment because of the dose escalation study (inferior disease control and fewer side effects) and photon SBRT had possible uncounted AEs because of the retrospectively collected data, this analysis showed similar OS, DSS, and LC rates. This study implies that there might not be a significant difference between CIRT and photon SBRT for the limited life expectancy group with localized small tumors. Thus, a prospective trial comparing SBRT with CIRT is mandatory to reveal genuinely effective indications for CIRT.

## Figures and Tables

**Figure 1 cancers-15-03633-f001:**
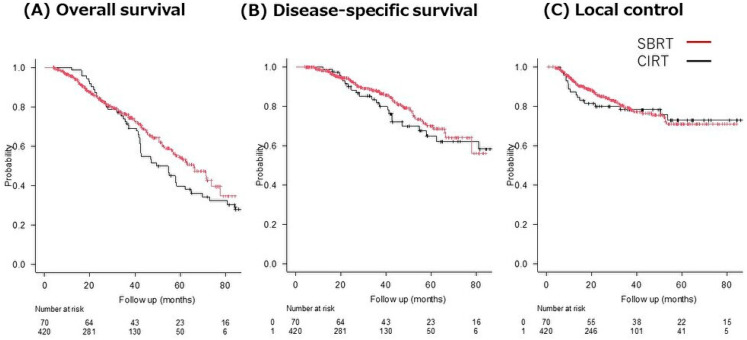
Overall survival (**A**), disease-specific survival (**B**), and local control (**C**) rates for CIRT and SBRT after radiation therapy.

**Table 1 cancers-15-03633-t001:** Patient characteristics.

		SRBT	CIRT	*p* Value
Total number		420	70	
Observation (months)	Median (range)	28.5 (3.9–84.2)	47.2 (12–128)	0.000024
Male/female		282/138	48/22	0.89
Median age (years)	median (range)	83 (80–93)	83 (80–89)	0.12
PS	0/1/2/unknown	169/196/43/12	36/26/4/0	
T1/T2 (UICC 6th)		279/141	39/31	0.072
Histology	adenocarcinoma	255	45	
	squamous	119	24	
	others	46	1	
Dose prescription (Gy/fr)	Median (range)	48 (40–70)/4–10	40 (28–50)/1	
(Gy (BED10))	Median (range)	105.6 (75–150)	200 (106.4–300)	

Abbreviations. SRBT, stereotactic body radiation therapy; CIRT, carbon-ion radiation therapy; PS, performance status; fr, fraction; BED, biologically effective dose.

## Data Availability

The data underlying this article will be shared upon reasonable request to the corresponding author.

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
