# Peer review of "Comparative Analysis of Photon Stereotactic Radiotherapy and Carbon-Ion Radiotherapy for Elderly Patients with Stage I Non-Small-Cell Lung Cancer: A Multicenter Retrospective Study"

_cancers, 2023, doi:10.3390/cancers15143633_

Round 1
Reviewer 1 Report
Manuscript ID: Cancers-2450783
Title: Comparative Analysis of Photon Stereotactic Radiotherapy and Carbon-Ion Radiotherapy for Elderly Patients with Stage I Non-small Cell Lung Cancer: A Multicenter Retrospective Study
Date: 2023/06/14
Reviewer's report:
This is an interesting manuscript as it’s a comprehensive analysis of photon SBRT vs carbon-ion RT for elderly patients with stage I NSLC. Surgery is recommended as the standard treatment for stage I non-small lung cancer (NSCLC), even in the elderly population . However, age-related mortality associated with pulmonary surgery increases with age .SBRT has already been established as an alternative standard treatment for inoperable or elderly stage I NSCLC. In recent years,treatment with carbon ions which provides several unique physical and radiobiologic properties have become one of the choices in the treatment for elderly patients with lung ca. This MS was one of the few study which thoroughly assessed the efficacy and safety of these two modalities (CIRT vs photon SBRT) and explored the differences between them. I'm sure this study could help in the decision-making process and guide towards an optimal management for early stage non-small cell ca of lung in elderly patients.
The MS is well prepared and containing a large amount of data. Although, there remain some limitation. Nevertheless, it was still well written. However, a few issue should be clarify prior publication.
1. How long does it take to complete 28-50 Gy (RBE) in 1 fraction of CIRT ? Was it longer than photon or proton SBRT ? Can elderly patients tolerate long treatment time ?
2. What CIRT treatment technique was used inorder to conform the dose distribution and to minimized dose to OAR ?
3. What was the tolerance dose for the various OARs ? Was it similar to proton therapy ?
4. How do you omit the fragmentation tail of CIRT ?
Author Response
Thank you for your attentive remarks.
Below, I will list the response.
- How long does it take to complete 28-50 Gy (RBE) in 1 fraction of CIRT? Was it longer than photon or proton SBRT? Can elderly patients tolerate long treatment time?
Beams using 2-4 ports are irradiated on the same day, and the time required to irradiate 50 Gy (RBE), the dose currently used, is about 30 minutes if there are no major positioning problems. This is the same as or rather shorter than SBRT using fixed multiport irradiation. We added these to the manuscript. (2.3. Radiation therapy)
- What CIRT treatment technique was used in order to conform the dose distribution and to minimized dose to OAR?
The key technology to minimize the dose to the OAR while covering the tumor thickness was the spread Bragg peak (SOBP) of the carbon ion beam with range modulators. We have added the above to the manuscript. (2.3. Radiation therapy)
- What was the tolerance dose for the various OARs? Was it similar to proton therapy?
The CIRT institution had its own restrictions on the dose of OARs. We have added the details. (2.3. Radiation therapy)
- How do you omit the fragmentation tail of CIRT?
The fragmentation tail of CIRT is not omit. Therefore, irradiation techniques such as respiratory-gated RT have been developed, and beam angles were selected to avoid t critical OARs and to reduce lung dose as much as possible.
Reviewer 2 Report
The manuscript reports the results of the retrospective clinical study aimed to compare two radiotherapy treatment modalities for non-small cell lung cancer – stereotactic body radiotherapy (SBRT) and carbon ion radiotherapy (CIRT). Efficacy and safety of treatment are two major criteria assessed in the study.
In overall, the subject of the study is of great interest for a wide audience of scientists and clinicians, from medical physicists and radiation oncologists to radiobiologists, and has both academic and clinical implications.
The authors concluded that two considered treatment modalities SBRT and CIRT were comparable in terms of efficacy, however CIRT was associated with lower toxicity. While these conclusions are justified by presented data based on the criteria selected for comparative analysis, some issues needs to be clarified.
The efficacy of treatment was assessed by three parameters – overall survival (OS), disease specific survival (DSS) and local control evaluated at 2- and 3-year intervals following treatment, for which no statistical difference was observed between SBRT and CIRT groups. Meanwhile, the visual inspection of time dependence for OS and DSS (presented in Figures 1) indicates quite substantial difference between SBRT and CIRT groups in the follow up interval between 40 and 80 months. This observation neither taken into consideration nor mentioned in the manuscript, thus leaving the reader with unanswered questions: are these differences statistically significant? are these differences important for drawing conclusions?
Author Response
Thank you for your attentive remarks.
Below, I will list the response.
- The authors concluded that two considered treatment modalities SBRT and CIRT were comparable in terms of efficacy, however CIRT was associated with lower toxicity. While these conclusions are justified by presented data based on the criteria selected for comparative analysis, some issues need to be clarified.
Thank you for your evaluation.
We would like to plan a toxicity analysis with more detailed data on OAR dosage and long-term follow-up in the future.
- The efficacy of treatment was assessed by three parameters – overall survival (OS), disease specific survival (DSS) and local control evaluated at 2- and 3-year intervals following treatment, for which no statistical difference was observed between SBRT and CIRT groups. Meanwhile, the visual inspection of time dependence for OS and DSS (presented in Figures 1) indicates quite substantial difference between SBRT and CIRT groups in the follow up interval between 40 and 80 months. This observation neither taken into consideration nor mentioned in the manuscript, thus leaving the reader with unanswered questions: are these differences statistically significant? are these differences important for drawing conclusions?
Thank you for pointing out the above. Unfortunately, the median observation period for this study was short: 28.5 months for the SBRT arm and 42.7 months for the CIRT arm. Also, the number of surviving cases in the CIRT arm at 40/60 months had decreased to 43/23 cases. Therefore, we do not see much value in comparing survival rates at each point after 3 years.
However, as you pointed out, we also believe it is necessary to conduct future studies with a sufficient observation period to provide more information, including long-term prognosis. We revised the "Limitations" section in part. (4.3. Limitations)